# Novel Mutation in *APC* Gene Associated with Multiple Osteomas in a Family and Review of Genotype-Phenotype Correlations of Extracolonic Manifestations in Gardner Syndrome

**DOI:** 10.3390/diagnostics11091560

**Published:** 2021-08-28

**Authors:** Cristina Antohi, Danisia Haba, Lavinia Caba, Mihai Liviu Ciofu, Vasile-Liviu Drug, Oana-Bogdana Bărboi, Bogdan Ionuț Dobrovăț, Monica-Cristina Pânzaru, Nicoleta Carmen Gorduza, Vasile Valeriu Lupu, Doina Dimofte, Cristina Gug, Eusebiu Vlad Gorduza

**Affiliations:** 1Odontology-Periodontology-Fixed Prosthetics Department, “Grigore T. Popa” University of Medicine and Pharmacy, 16 University Street, 700115 Iasi, Romania; crisantohi_med@yahoo.com; 2Oral and Maxillofacial Surgery Department, “Grigore T. Popa” University of Medicine and Pharmacy, 16 University Street, 700115 Iaşi, Romania; danihaba@yahoo.com (D.H.); bodan.dobrovat@yahoo.com (B.I.D.); 3Medicine of Mother and Child Department, “Grigore T. Popa” University of Medicine and Pharmacy, 16 University Street, 700115 Iasi, Romania; monica.panzaru@yahoo.com (M.-C.P.); valeriulupu@yahoo.com (V.V.L.); vgord@mail.com (E.V.G.); 4Medical I Department, “Grigore T. Popa” University of Medicine and Pharmacy, 16 University Street, 700115 Iasi, Romania; vasidrug@email.com (V.-L.D.); oany_leo@yahoo.com (O.-B.B.); 5Endocrinology Department, “St. Spiridon” Hospital, 700111 Iasi, Romania; cargorduza@yahoo.fr; 6Medoptica, 700194 Iasi, Romania; doinadimofte@yahoo.com; 7Microscopic Morphology Department, “Victor Babes” University of Medicine and Pharmacy, 300041 Timișoara, Romania; dr.cristina.gug@gmail.com

**Keywords:** familial adenomatous polyposis, hereditary, osteomas, genomic variant

## Abstract

Gardner syndrome is a neoplasic disease that associates intestinal polyposis and colorectal adenocarcinoma with osteomas and soft tissue tumors determined by germline mutations in the *APC* gene. The early diagnosis and identification of high-risk individuals are important because patients have a 100% risk of colon cancer. We present the case of a family with Gardner syndrome. Cephalometric, panoramic X-rays and CBCT of the proband and her brother showed multiple osteomas affecting the skull bones, mandible and paranasal sinuses. The detailed family history showed an autosomal dominant transmission with the presence of the disease in the mother and maternal grandfather of the proband. Both had the typical signs of disease and died in the fourth decade of life. Based on these aspects the clinical diagnosis was Gardner syndrome. By gene sequencing, a novel pathogenic variant c.4609dup (p.Thr1537Asnfs*7) in heterozygous status was identified in the *APC* gene in both siblings. We reviewed literature data concerning the correlation between the localization of mutations in the *APC* gene and the extracolonic manifestations of familial adenomatous polyposis as well as their importance in early diagnosis and adequate oncological survey of patients and families based on abnormal genomic variants.

## 1. Introduction

Colorectal cancer (CRC) is the third most common cancer worldwide (approximately 1.8 million new cancer diagnoses yearly) and the second most common cause of cancer death (over 880,000 deaths in 2018) [1]. A special entity is represented by CRC with early onset that has a different clinical, pathological and molecular profile compared to CRC with a late onset. Thus, distinction between the two forms is important for effective prevention, detection and the therapeutic plan [2].

Usually, colorectal cancer in young people is discovered at an advanced stage due to the lack of screening programs in this age group. Exceptions are cases of hereditary syndromes predisposed to CRC (10% of all colorectal cancers), where, after the discovery of an index case, a large screening of the family follows [3,4].

Most early-onset cancers are monogenic (16–35% of total) [2,4,5,6,7]. The most common forms are Lynch Syndrome and familial adenomatous polyposis (FAP). The high percentage of monogenic cancers with early onset indicates and justifies genetic counselling and multigene panel testing [4]. However, nearly half of patients with a genetic form of CRC do not have a positive family history [6]. Thus, genetic diagnosis is important for patients, but also for the identification and management of relatives at risk of inheriting the mutation.

Familial adenomatous polyposis accounts for approximately 1% of all colorectal cancers [8]. FAP (OMIM #175100) is caused by germline mutations in a tumor suppressor gene—*APC* regulator of WNT signaling pathway (*APC*). Mutations of the *APC* gene have an autosomal dominant inheritance pattern. FAP can be divided in several forms: classic FAP (characterized by the presence of a large number >100 of colorectal adenomatous polyps), attenuated FAP (AFAP—characterized by a lower number of polyps, usually under 30), gastric adenocarcinoma and proximal polyposis of the stomach (GAPPS), Gardner syndrome (associates FAP with osteomas and soft tissue tumors) and Turcot syndrome (association of FAP with central nervous system tumors) [9].

The presence of extracolonic non-malignant manifestations in Gardner syndrome (including osteomas, dental abnormalities, congenital hypertrophy of the retinal pigment epithelium, benign cutaneous lesions, desmoid tumors and adrenal masses) and their early appearance, even before the development of polyps/cancer, is an important factor for diagnosis and prevention.

We present two cases, siblings diagnosed with a new pathogenic variant in the *APC* gene to highlight the importance of extracolonic manifestations and family history in the early diagnosis of FAP. We also reviewed the extracolonic manifestations in FAP and their correlations with the location of mutations in the *APC* gene.

## 2. Materials and Methods

### 2.1. Patients

Two patients were included in the study: the proband and her brother. The proband is a young Caucasian female (16 year old at the time of diagnosis). She was referred to an oral surgeon because of a mandibular osteoma discovered during a routine consultation. The proband’s brother (12 year old at the time) had supernumerary teeth and a mandibular tumor on palpation. Family history showed that the proband’s mother was diagnosed with mandibular osteoma, supernumerary teeth and died at 34 years old because of intra-abdominal and abdominal wall tumors. The maternal grandfather had supernumerary teeth and soft tissue tumors and died around the age of 35 (Figure 1).

The association of multiple osteomas, dental abnormalities, and family history of colon cancer with dominant transmission has led to clinical diagnosis of familial adenomatous polyposis Gardner variant.

### 2.2. Methods

Patients were evaluated by a multidisciplinary team consisting of an oral surgeon, radiologist, geneticist, gastroenterologist, endocrinologist and an ophthalmologist.

A complete evaluation was performed by anterior-posterior cephalometric and panoramic X-rays, Cone-beam computed tomography (CBCT), digestive endoscopic evaluation, molecular testing, ophthalmological and endocrinological evaluation. At the age of 18, the proband underwent surgical removal of the most prominent osteoma of the left angle of the mandible and a pathological examination was performed.

The patients underwent Invitae Multi-Cancer panel (Invitae, San Francisco, CA, USA). Full-gene sequencing, deletion/duplication analysis and variant interpretation were performed at Invitae (Appendix A).

## 3. Results

### 3.1. Radiographic Assessment

Initial radiographic assessment of the female proband (III.1.) included: anterior-posterior cephalometric and panoramic X-rays which showed multiple osteomas affecting the skull bones and mandible, slight facial asymmetry and several impacted teeth (Figure 2a–c). CBCT revealed diffuse sclerosis of the jaw bones (Figure 2d–g), expansion of the body of the mandible, multiple peripheral osteomas including a large osteoma at the left angle of the mandible associated with facial asymmetry (Figure 2d–j). Small osteomas were registered within the anterior ethmoidal cells, frontal sinus, several skull bones (frontal, temporal, parietal, occipital), zygomatic arch, nasal bones (Figure 2g–j) and the left external auditory canal with a discrete decrease of its diameter (Figure 2i).

Cephalometric and panoramic radiographs of the male patient (III.2.) showing multiple radiopaque masses with relatively well-defined borders affecting the jaw bones, diffuse sclerosis of the body of the mandible and large masses near the gonions (Figure 3a,b). CBCT axial, cross-section, panoramic, sagittal, coronal, 3D lateral-oblique and 3D sagittal reconstructions showing multiple enostoses located in the medullary bone of the body of the maxilla and mandible, delayed dental eruptions and hypercementosis (Figure 3c–i). Bilateral mushroom-shaped osteomas affect the angle of the mandible, temporal and frontal bones (Figure 3h,i).

In comparison, the male patient presented overall larger osteomas affecting the left angle of the mandible and both temporal bones, left coronoid process, impacted teeth, and increased bone sclerosis of the jaw bones. Additionally, a breast lipoma was observed in the male patient. Specific for this case is the presence of a pedunculated osteoma attached to the anterior wall of the right sphenoidal sinus and mucositis of the posterior ethmoidal cells.

### 3.2. Pathological Examination

The pathological examination showed compact bone tissue, consisting of mature bone lamellae with Haversian canals and paucicellular fibrous stroma (Figure 4). These morphological aspects confirm the presence of osteoma.

### 3.3. Digestive Endoscopic Evaluation

Upper digestive endoscopy in the proband showed a normal aspect of esophageal and gastric mucosa. A pseudopolipoid appearance (<2 mm) and hypertrophic papilla were observed in the duodenum. Lower digestive endoscopy revealed multiple polyps (approximately 50–70), the largest located in the sigmoid colon with a diameter of 5–6 mm. Polypoid lesions were observed in cecum (a 2 mm polyp) and at 5 cm from the ileocecal valve (also a 2 mm polyp).

Pathological examination of the biopsied fragments shows the structure of tubular adenomatous polyp with low-grade epithelial dysplasia. The colonic mucosa fragment has glandular architecture and preserved mucosecretion, with moderate lymphoplasmacytic inflammatory infiltrate and chorion edema.

Similar aspects of colon were discovered by endoscopy in the male patient.

### 3.4. Genetic Testing

Gene sequencing (by NGS) showed a novel pathogenic variant—c.4609dup (p.Thr1537Asnfs*7)—in heterozygous status in the *APC* gene in the siblings. The reference sequence is: *APC*: NM_000038.5. The novel sequence has a premature translational stop signal in the *APC* gene (p.Thr1537Asnfs*7). While this is not anticipated to result in nonsense mediated decay, it is expected to disrupt the last 1307 amino acids of the APC protein. This variant is not present in population databases (ExAC no frequency) [10]. It is expected to disrupt the domains of the protein, which mediate interactions with the cytoskeleton. A different truncation (p.Tyr2645Lysfs*14) that lies downstream of this variant has been determined to be pathogenic. This suggests that deletion of this region of the APC protein is causative of disease. For these reasons, this variant has been classified as pathogenic [10]. No pathogenic variants were identified in the other genes analyzed.

Ophthalmologic and endocrine examinations did not reveal particular aspects concordant with diagnosis of Gardner syndrome.

The diagnosis of Gardner syndrome was established based on the presence of multiple adenomatous polyps in the colon, the presence of extracolonic manifestations (multiple osteomas, dental abnormalities, soft tissue tumor), suggestive familial history for FAP, being confirmed by the discovery of a pathogenic variant in the *APC* gene.

So far, the female patient underwent surgical removal of the mandibular osteoma and orthodontic treatment. The male patient benefitted only from orthodontic treatment due to incomplete ossification. Both patients entered the surveillance oncological program with an annual endoscopic evaluation of the digestive mucosa and biopsy of the lesions.

## 4. Discussion

The APC protein is a complex protein, having several domains with different cellular roles. The APC protein contains an oligomerization domain; an armadillo repeat-domain (implied in cell migration and cell adhesion by link of Asef, KAP3 and IQGAP1); a 15–20 aa repeats that fix β-catenin and participates in cell adhesion, proliferation and differentiation; SAM repeats that allow connection with axin; a basic region for microtubule binding and C-terminal domains for EB1 binding. The last two domains are implied in chromosomal segregation and mitotic progression [11].

The majority of mutations are missense or frameshift, produce a truncated protein and modify the last exon that contains 75% of the coding sequence of the gene [12,13]. The region between codons 1250 and 1464 is considered a mutation cluster region (MCR) and mutations of this region have a higher frequency in the general population (15.49% of total) [14]. The significant consequences seem to be mutations in domains implied in binding β-catenin, EB1 and microtubules. These mutations disrupt cellular processes and generate the appearance of tumors by stimulation of cell migration, activation of proliferation and inhibition of differentiation. Additionally, chromosomal instability and cell immortalization are produced [11]. The most common mutations are in codon 1309 (10% of patients with FAP) and codon 1061 (5% of patients); 25–30% of mutations are de novo. In these cases, there is no evidence of FAP phenotype or mutations in the *APC* gene in family members. In some cases, the mutation exists only in the gametes (germinal mosaicism) [15].

The types of mutations in the *APC* gene included in the Human Gene Mutation Database (HGMD^®^) are: missense/nonsense (578 variants), splicing substitutions (125 variants), regulatory substitutions (13 variants), small deletions (20 pb or less; 796 variants), small insertions/duplications (20 pb or less; 343 variants), small indels (20 pb or less; 51 variants), gross deletions (143 variants), gross insertions/duplications (16 variants), complex rearrangements (13 variants) (HGMD^®^ Professional 2021.1, accessed on 26 June 2021) [16]. The distribution of germline mutations among exons is presented in Figure 5.

The gene panel used for our patients contains genes involved in hereditary polyposis syndromes, some of which are relatively new but also genes involved in other types of hereditary cancers. Using such a panel offers accurate differential diagnosis with the intestinal polyposis syndromes. Among those genes, we mention: *NTHL1* (nth like DNA glycosylase 1), *POLE* (DNA polymerase epsilon, catalytic subunit), *POLD1* (DNA polymerase delta 1, catalytic subunit), *MSH3* (mutS homolog 3), *AXIN2* (axin 2), *GREM1* (gremlin 1, DAN family BMP antagonist), *MUTYH* (mutY DNA glycosylase), *STK11* (serine/threonine kinase 11), *SMAD4* (SMAD family member 4) and *BMPR1A* (bone morphogenetic protein receptor type 1A). All these genes are found in the panel of genes recommended for testing for inherited colorectal cancer and polyposis by the American College of Medical Genetics and Genomics [9]. The use of targeted gene panel-based next-generation sequencing technology is recommended for patients with FAP because it is cost effective, and less time consuming. This approach is more targeted but delivers less information than whole genome sequencing (WGS) as only a subset of genes is evaluated [17].

### 4.1. Extracolonic Manifestations in FAP

Penetrance is almost complete for colonic manifestations, but variable for extra colonic manifestations. The development of adenomatous polyps begins in childhood and adolescence. It is estimated that 50% of patients with FAP have colorectal adenomas by age 15 and the percentage increases to 95% by age 25 [18]. The age of onset of colonic symptoms correlates with the location of the mutation: age 20 (variants at 1309 codon), age 30 (variants between 168 and 1580 codons except for the 1309 codon), age 52 (variants in 5′ of codon 168 and 3′ of codon 1580) [19].

There are two categories of extracolonic manifestations in FAP: malignant and non-malignant. They are present in 70% of FAP cases [20]. Non-malignant extra intestinal manifestations are represented by: osteomas, dental abnormalities, congenital hypertrophy of the retinal pigment epithelium (CHRPE), benign cutaneous lesions, desmoid tumors and adrenal masses [21]. Malignant manifestations are located in the small bowel (duodenum, periampulla or distal to the duodenum), pancreas, thyroid, CNS, liver, bile ducts and stomach [21].

### 4.2. Osteomas

Osteomas are present in 60–80% of patients with FAP and 50% of all osteomas occur in FAP [22,23]. The prevalence of osteomas is 1–2% in the general population [24]. Osteomas are benign tumors characterized by compact lamellar cortical or cancellous bone [25]. There are three types of osteomas: central (originates in the endosteum), peripheral (originates in the periosteum) and extra skeletal soft tissue osteoma (developing in the muscles) [26]. They are most commonly located in the skull (jaw bones and paranasal sinuses), rarely in long bones or muscles. Osteomas are more common in the mandible than in the maxilla [27].

In descending order of the frequency of osteomas at the level of the sinuses, we mention: frontal sinus, ethmoid, maxillary and sphenoid sinuses. The prevalence in the paranasal sinuses is different depending on the imaging method used for their detection: conventional examination shows a prevalence of about 1% in the general population, while the use of computed tomography has a higher detection rate showing a prevalence of about 3% in the general population [28]. The presence of multiple osteomas (more than three osteomas) is suggestive for Gardner syndrome [28,29]. In a prospective study of paranasal osteomas, Erdogan et al. showed that in 7% of cases, osteomas were detected in more than one sinus [28].

Osteomas are important in the early management of FAP due to the fact that they occur before intestinal manifestations—the detection of osteomas precedes the diagnosis of FAP by 17 years [23]. The appearance of osteomas is associated with mutations between codons 767 and 1578 [24]. In our case the mutation is present in the same region.

Another particularity of our patients is the presence of multiple osteomas in cranial bones, jaw bones and paranasal sinuses. However, osteoma of sphenoid sinus (present in patient III.2) is a less common location.

### 4.3. Dental Abnormalities

Dental abnormalities are present in 30–75% of patients with FAP [30]. The meta-analysis performed by Almeida et al. in 2015 shows a frequency of dental anomalies of 30.48% [31]. Dental abnormalities in FAP are represented by: impacted teeth, congenitally missing teeth, supernumerary teeth, dentiferous cysts, compound odontomas, hypercementosis [21,22]. Supernumerary teeth were identified in 11–27% of FAP patients compared to 0–4% in the general population. The most common location is anterior and around canines, in the alveolar bone between the teeth or attached to the follicle of an impacted tooth. However, this location is not specific to FAP [32,33]. The literature is poor in studies about the correlation between the presence of dental abnormalities and the location of mutations in the *APC* gene. It was assumed that there was the same correlation as in the case of desmoid tumors and osteomas [34,35].

Due to the existence of dental abnormalities, a dentist may be involved in the early diagnosis of familial adenomatous polyposis.

Supernumerary teeth can be an important clue for early diagnosis. We found dental modifications in our cases. The two siblings have impacted teeth and hypercementosis, while their mother and their maternal grandfather showed supernumerary teeth.

### 4.4. Desmoid Tumors

Desmoid tumors (DT) appear in the musculoaponeurotic tissue in any region of the body. There are 3 common locations: extremities (proximal or at the girdle), abdominal wall and intra-abdominal (intestinal wall and mesentery) [36]. They represent 3% of soft tissue tumors and 0.03% of all neoplasms [36]. DT occurs in 12–15% of FAP patients [37]. They generate a high mortality in patients with FAP, with a rate between 10% and 50% [38]. The patient with FAP has an 800-fold higher risk of developing DT and the risk increases if there is a mutation after exon 1399 in the *APC* gene [37]. DT in FAP appear earlier than sporadic DT [39]. In 80% of patients with DT, they appear until the age of 40 with a peak in the second and third decade of life [37]; 65% of DT occur intra-abdominally [37]. There are four risk factors for DT: a positive family history for DT, mutation location, surgical trauma, female sex [37].

The correlation between the appearance of DT and the location of the mutation is not highly specific. Friedl et al. found a 60% frequency of DT in patients with FAP caused by mutations in codons 1445–1580, but a 20% frequency of DT in patients with mutations before codon 1444 in the FAP gene [19].

### 4.5. Congenital Hypertrophy of the Retinal Pigment Epithelium

Congenital hypertrophy of the retinal pigment epithelium (CHRPE) is a pigmented lesion with a depigmented halo in the retina and the most common extracolonic manifestation found in 74% of FAP patients compared to 1.2–4.4% in the general population [40,41]. It has no clinical significance and it is asymptomatic. It is the earliest extracolonic sign present at birth or in the neonatal period [41]. It is associated with mutations in the codon region 311–1444 [42]; 70.74% of the mutations described in HGMD are in this codon range.

### 4.6. Malignant Extracolonic Manifestations

Malignant extracolonic manifestations in FAP are represented by an additional cancer in addition to colon cancer in FAP. Lifetime risk for cancer in FAP differs among cancer types [21]. Some entities are rare, such as small bowel cancer distal to the duodenum, others are more common, such as thyroid cancer or small bowel cancer (duodenum or periampulla) (Table 1). Studies concerning this type of cancer are rare, in correlation with their small frequency.

Mutations in the APC gene 697–1224 codons are associated with a 3-fold higher risk of brain tumor and a 13-fold higher risk of medulloblastoma [43]. Duodenal adenomas have a 3–4 times higher risk of developing if the mutation in *APC* is between codons 976–1067 [44].

### 4.7. Thyroid Cancer

The incidence of thyroid cancer is 2–12% in cases with FAP and 10% in the Gardner variant with a cumulative risk of 2.8% by age 60 [45]. This neoplasia is multifocal and bilateral. In cases with FAP, the majority of cases are cribriform morular variant of papillary CT [46]. It is associated with germline mutations between the codons 1286 and 1513 in the *APC* gene [45,47]. In the respective codons interval, there are 251 variants which represent 14.40% of the total different mutations reported in HGMD up to date. It predominates in women and is diagnosed earlier than sporadic with an average diagnostic age of 29.2 years [47,48,49]. In a systematic review, Septer et al. concluded that there is an increased risk of thyroid cancer associated with a mutation in codon 1061 [49].

## 5. Conclusions

The genetic syndromes predisposed to colorectal cancer (including FAP) are characterized by pleiotropy and genetic heterogeneity. Mutations in the *APC* gene, which later lead to familial adenomatous polyposis manifest earlier as osteomas, dental abnormalities, congenital hypertrophy of the retinal pigment epithelium, benign cutaneous lesions, desmoid tumors and adrenal masses. An early diagnosis via these early manifestations allows a proper oncological surveillance. Furthermore, a detailed family history in patients with these extracolonic manifestations is very important in identifying the family members at risk.

## Figures and Tables

**Figure 1 diagnostics-11-01560-f001:**
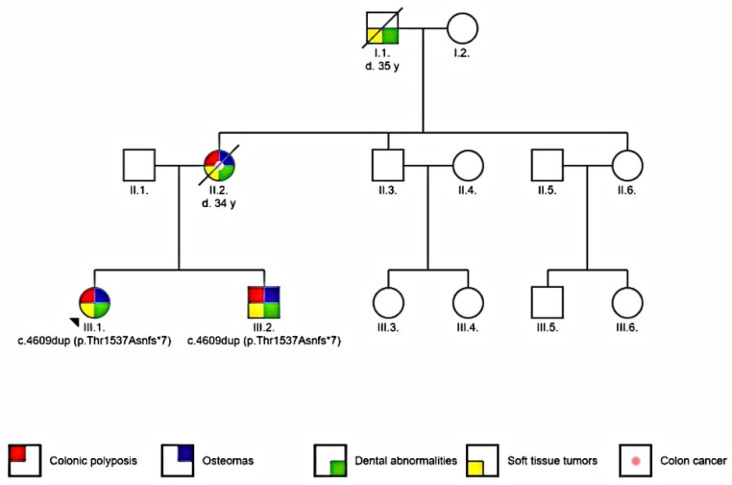
Pedigree of the analyzed family (arrow indicates the proband).

**Figure 2 diagnostics-11-01560-f002:**
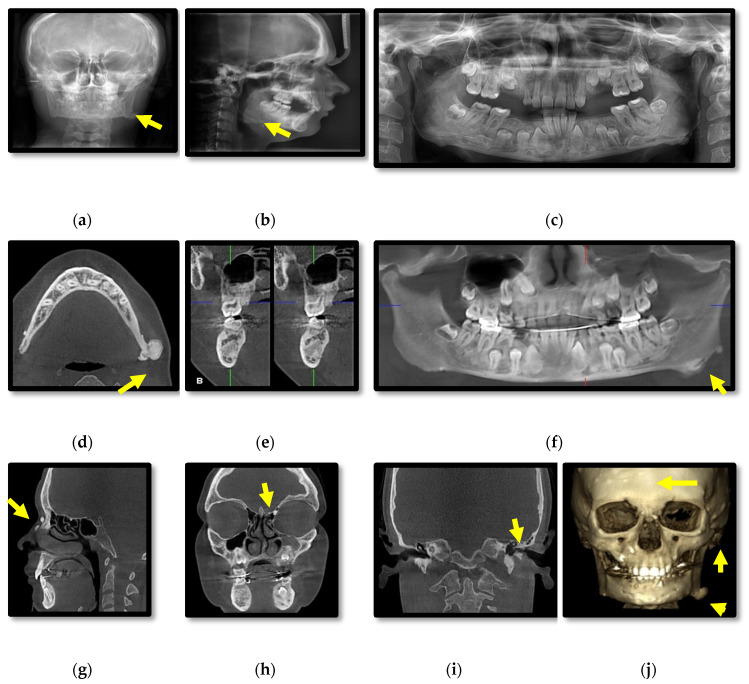
Patient III.1. Cephalometric (**a**,**b**) and panoramic (**c**) radiographs; Cone-beam computed tomography (CBCT) with axial (**d**), cross-section (**e**), panoramic (**f**), sagittal (**g**), coronal (**h**,**i**) views and 3D reconstructions of bone tissue (**j**).

**Figure 3 diagnostics-11-01560-f003:**
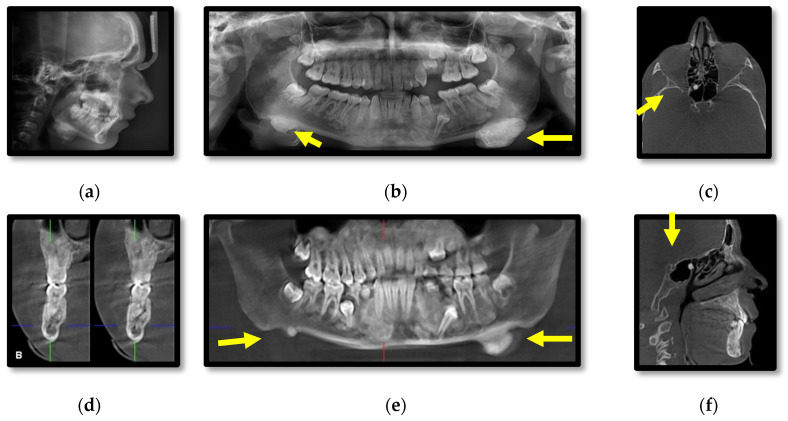
Patient III.2. Cephalometric (**a**) and panoramic (**b**) radiographs; CBCT axial (**c**), cross-section (**d**), panoramic (**e**), sagittal (**f**), coronal (**g**), 3D lateral-oblique (**h**) and 3D sagittal (**i**).

**Figure 4 diagnostics-11-01560-f004:**
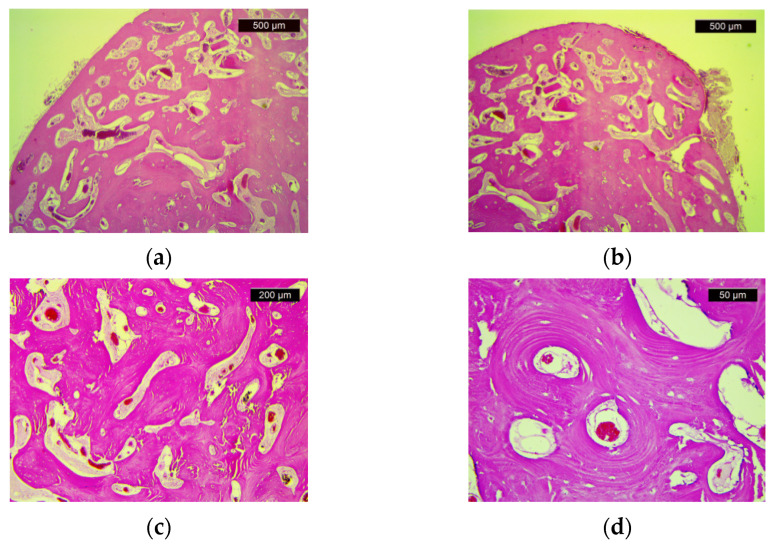
Histologic features of osteoma: (**a**) Dense cortical lamellar bone tissue (HE, ×25); (**b**) Dense, compact bone tissue with marked paucicellular and congestive stromal fibrosis (HE, ×25); (**c**) Compact, dense bone with visible Haversian canals (HE, ×50); (**d**) Haversian canals (HE, ×200).

**Figure 5 diagnostics-11-01560-f005:**
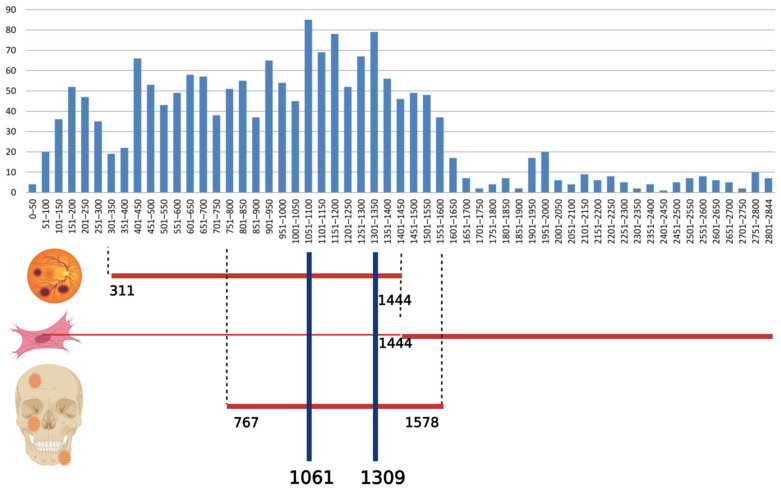
Distribution of germline mutations in *APC* gene (data from the online database HGMD^®^ Professional 2021.1, accessed on 26 June 2021) and correlation with congenital hypertrophy of the retinal pigment epithelium, desmoid tumors and osteomas. The abscissa represents the APC codon number and the ordinate represents the number of different mutations. Created with BioRender.com.

**Table 1 diagnostics-11-01560-t001:** Extracolonic cancer in FAP—lifetime risk (modified after *Jasperson*) [21].

Extracolonic Cancer	Lifetime Risk for Cancer Observations
small bowel (duodenum or periampulla) cancer	4–12%
small bowel (distal to the duodenum) cancer	rare
pancreas cancer	1%
thyroid cancer	1–12%
CNS cancer	<1%
liver cancer	1.6%
bile ducts cancer	Low, but increased
stomach cancer	<1%

## Data Availability

Some data were obtained from Human Gene Mutation Database (HGMD^®^ Professional 2021.1, accessed on 26 June 2021) available online: http://www.hgmd.cf.ac.uk/ac/all.php.

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
