# Peer review of "Novel Mutation in APC Gene Associated with Multiple Osteomas in a Family and Review of Genotype-Phenotype Correlations of Extracolonic Manifestations in Gardner Syndrome"

_diagnostics, 2021, doi:10.3390/diagnostics11091560_

Round 1
Reviewer 1 Report
General: The article is a detailed case study with an extensive review of the relevant literature. Unfortunately the language is rather bad; at times I did not get what the authors meant. In the beginning, the authors talk about a sister/brother sibling pair; later they use the term "brothers" to describe their cases. Such language problems (I assume it is only language) distract from the scientific content. In particular, the section "Conclusions" is hard to read and I had to guess the meaning. I suggested more intelligable phrasing, but may have missed the meaning. As minor annoyance, the use of articles is almost consistently wrong. The article would profit from extensive language editing. (I add a commented manuscript, which may help the authors. Please be aware that I did not aim to correct the whole manuscript.)
I am a geneticist by training and also have a background in statistics. Hence I may not represent the medical side adequately.
With respect to the content: a mutation in the APC gene not yet identified before seems to lead to early onset colon cancer. Inheritance seems to be dominant: the mother and grandmother seem to have suffered from the same syndrome. Before the colorectal cancer manifests, craniofacial and dental malformations were evident. These malformations may provide a basis for early diagnosis and allow for early oncological screening even before the deadly CRC sets in.

Author Response
Thank you very much for your comments and constructive observations.
We corrected the suggested grammatical mistakes (lines 24, 29, 30, 31, 32, 33, 45, 53, 55, 154, 162, 184, 207, 271, 315)
Line 160: we used “siblings”
Line 174: we deleted the subtitle
Line 191: we explained what means “germinal mosaicism”
Line 225: we replaced “without” with “except the”
Line 256: we replaced “other” with “another”
Line 274: we replaced “brothers” with siblings
Line 301: we replaced the phrase with “Malignant extracolonic manifestations in FAP are represented by an additional cancer in addition to colon cancer in FAP”
Line 302: we have entered the corresponding bibliographic reference
Line 321,322,324,327: We rewrote the conclusions:
“The genetic syndromes predisposed to colorectal cancer (including FAP) are characterized by pleiotropy and genetic heterogeneity. Mutations in the APC gene, which later lead to familial adenomatous polyposis manifest earlier as osteomas, dental abnormalities, congenital hypertrophy of the retinal pigment epithelium, benign cutaneous lesions, desmoid tumours and adrenal masses. An early diagnosis via these early manifestations allows a proper oncological surveillance. Furthermore, a detailed family history in patients with these extracolonic manifestations is very important in identifying the family risk members.”
The manuscript was verified for language mistakes and rectified accordingly.
Reviewer 2 Report
Having read the review my initial impressions are that it is well written and the clinical aspects of it are well presented. Somehow though the Next generation sequencing component which explores a panel of genes associated with cancer is not very well integrated and the rational for their inclusion is not clear. The APC gene is presented as the causative gene but other data on the remaining genes that were genotyped is not presented or explained.
The description in the analysis is not adequate nor is the follow up of the 143 genes in terms of why they were genotyped.
"The patients underwent a comprehensive multigene panel analysis of 143 genes implied in hereditary cancer. The genetic test allowed sequences analysis and deletion/duplication investigation. Genomic DNA obtained are enriched for the targeted regions using a hybridization-based protocol and after that sequenced using Illumina® technology"
ln 159 - the authors identify a novel genetic variant but if this is completely novel how can its pathogenicity be established in a cohort of two. Also, there is no accession number or RS number associated with the variant. This variant need to be put in context with others.
Gene sequencing (by NGS) showed a novel pathogenic variant - c.4609dup (p.Thr1537Asnfs*7) - in heterozygous status in APC gene in the two brothers. The reference sequence is: APC: NM_000038.5.
There are some grammatical errors throughout and other comments which I have highlighted in the attached document.

Author Response
Thank you very much for your comments and constructive observations.
Line 46: we replaced “tends to be” with “is”
Line 50: we replaced “a big amount” with “Most”
Line 55: we added the article “the” for “patient” and “identification”
Line 59: we deleted “these”
Line 60: we added “pattern”
Line 67: we added “including”
Line 68: we replaced “theirs” with “their”
Line 72: we deleted “in order”
Line 91: we replaced “in” with “within”
Line 97: we replaced “gonion” with “angle of the mandible”
Line 99: we replaced “sequences” with “sequence”
Line 100: we replaced “are” with “were”
Line 101: we added the method “Next-Generation Sequencing”
Line 159: we explained that this variant is not present in population databases (ExAC no frequency). We also explained why it was considered a pathogenic variant according to the references in ClinVar (reference with accession number: National Center for Biotechnology Information. ClinVar; [VCV000858116.2], https://www.ncbi.nlm.nih.gov/clinvar/variation/VCV000858116.2 (accessed Aug. 8, 2021).
“This sequence change results in a premature translational stop signal in the APC gene (p.Thr1537Asnfs*7). While this is not anticipated to result in nonsense mediated decay, it is expected to disrupt the last 1307 aminoacids of the APC protein. This variant is not present in population databases (ExAC no frequency) [10]. This variant is expected to disrupt the domains of the protein, which mediate interactions with the cytoskeleton. A different truncation (p.Tyr2645Lysfs*14) that lies downstream of this variant has been determined to be pathogenic. This suggests that deletion of this region of the APC protein is causative of disease. For these reasons, this variant has been classified as Pathogenic [10]. No
pathogenic variants were identified in the other genes analysed.”
We would like to highlight the fact that ClinVar mentions: “This variant has been observed in a family with clinical features of familial adenomatous polyposis (PMID: 19029688)”. In this article (PMID: 19029688 - Plawski, A. and Slomski, R., 2008. APC gene mutations causing familial adenomatous polyposis in Polish patients. Journal of applied genetics, 49(4), pp.407-414, doi: 10.1007/BF03195640) a mutation in codon 4609 is described, but it is a deletion and not a duplication as in our case.
Line 174: we deleted the subtitle
Line 216: we replaced “has good cost-effectiveness” with “is cost effective”
Line 218: we rephrased: “This approach is more targeted but delivers less information than whole genome sequencing (WGS) as only a subset of genes is evaluated”
The manuscript was verified for language mistakes and rectified accordingly.
Round 2
Reviewer 1 Report
The manuscript is much easier to read now. I still have some minor language comments (see my comments in the attached manuscript). I doubt that I caught all mistakes.

Author Response
We thank the reviewer for giving us the opportunity to improve the quality of the manuscript.
Line 71: We deleted “of”
Line 78: We introduced “the”
Line 79: We introduced “the”
Line 82: We introduced “the”
Line 92: We replaced “within” with “by”
Line 116: We deleted ‘
Line 164: We introduced “the”
Line 165: We replaced “This” with “The novel”
We replaced “change results in” with “has”
Line 169: We replaced “This variant” with “It”
Line 173: We replaced “P” with “p”
Line 175: We replaced “revealed” with “reveal”
Line 188: We introduced “the”
Line 197: We replaced “the changing” with “mutations”
Line 198: We replaced “are the” with “seem to be”
We replaced “that interest” with “in”
Line 204: We replaced “are” with “is”
Line 207: We introduced “the”
We replaced “includes” with “included”
Line 221: We deleted “an”
Line 253: We replaced “3” with “three”
Line 300: We replaced “te” with “the”
Line 302: We replaced “4” with “four”
Line 304: We replaced the phrase with “The correlation between the appearance of DT and the location of the mutation is not highly specific”
Line 318: We replaced “is different” with “differs among cancer types”
Reviewer 2 Report
There are still some minor typographical errors outstanding (highlighted in document), mainly to do with formatting, i.e. spacing ect. One thing that is outstanding is that line 100-104 does not adequately describe the method for genotyping, the part number for the gene panel (nor the gene panel itself)https://www.invitae.com/en/physician/tests/01101/ are described, this should be listed into the supplementary materials if it is a custom panel or provided as a part number if it is off the shelf. Also the full address for Illumina is not provided

Author Response
We thank the reviewer for giving us the opportunity to improve the quality of the manuscript.
Line 33: We wrote “both siblings”
Line 49: We removed the extra space
Line 165: We changed
Line 168: We replaced “aminoacids” with “amino acids”
Line 231: We removed the extra space
Line 240: We removed the extra space
We completed with Illumina address
We made a supplementary material (Supplementary Materials) with genes of the panel and some considerations concerning the methods used.